# MCLF: A Multi-grained Contrastive Learning Framework for ASR-robust Spoken Language Understanding

**Zhiqi Huang, Dongsheng Chen, Zhihong Zhu, and Xuxin Cheng**

Peking University, China

zhiqihuang@pku.edu.cn

{chends,zhihongzhu,chengxx}@stu.pku.edu.cn

## Abstract

Enhancing the robustness towards Automatic Speech Recognition (ASR) errors is of great importance for Spoken Language Understanding (SLU). Trending ASR-robust SLU systems have witnessed impressive improvements through global contrastive learning. However, although most ASR errors occur only at local positions of utterances, they can easily lead to severe semantic changes, and utterance-level classification or comparison is difficult to distinguish such differences. To address the problem, we propose a two-stage multi-grained contrastive learning framework dubbed MCLF. Technically, we first adapt the pre-trained language models to downstream SLU datasets via the proposed multi-grained contrastive learning objective and then fine-tune it on the corresponding dataset. Besides, to facilitate contrastive learning in the pre-training stage, we explore several data augmentation methods to expand the training data. Experimental results and detailed analyses on four datasets and four BERT-like backbone models demonstrate the effectiveness of our approach.

## 1 Introduction

The rise in popularity of intelligent assistants like Apple Siri and Amazon Alexa has aroused interest in intelligent speech technology. A fundamental component of these smart assistants is the Spoken Language Understanding (SLU), a framework that aims to comprehend the semantics of human speech (Tur and Mori, 2011). Traditionally, semantic information from utterances are obtained by initially applying Automatic Speech Recognition (ASR) to convert human speech into text. Afterwards, the SLU model is trained using this textual transcript as input (Huang et al., 2021c; Chen et al., 2022a; Zhu et al., 2023c,b).

Since speech-to-text has a wide range of application scenarios in both academia and industry, many ASR systems based on neural networks have

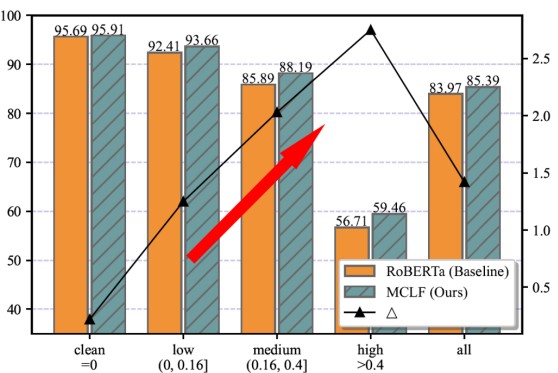

Figure 1: Comparison of RoBERTa$_{base}$ and our MCLF on different WER intervals split from the SLURP test set. The performance of MCLF in the case of larger WER is greater than that of RoBERTa$_{base}$.

been proposed, *e.g.*, Speech Transformer (Dong et al., 2018), wav2vec (Baevski et al., 2020), and SPIRAL (Huang et al., 2022a). Despite the success, ASR systems still generate transcripts with errors (mainly *insertions*, *deletions*, and *substitutions* errors) when facing more complicated real-world scenarios (Fang et al., 2020a). These ASR errors may confuse the downstream SLU tasks and lead to performance degradation. Previous works towards ASR-robust SLU can be divided into two categories: 1) Detecting and correcting ASR errors directly (Zhou et al., 2022), or injecting additional ASR output information such as N-best hypothesis and phoneme sequence to the downstream SLU models (Li et al., 2020; Sundararaman et al., 2021; Wang et al., 2022b). However, both hypotheses with ASR errors and extra ASR information are hard to obtain due to the limitation of the ASR system (Dutta et al., 2022). 2) Adapting the ASR system architecture towards improving language correctness and fluency (Kim et al., 2021a; Le et al., 2021). An obvious shortcoming is the additional technical burden of modifying and retraining existing ASR systems according to specific tasks.

Recent pre-trained language models (PLMs) like

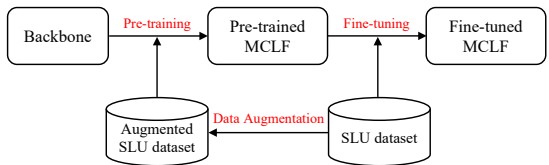

Figure 2: Overview of MCLF learning.

BERT (Devlin et al., 2019; Liu et al., 2019b) have shown appealing performance on various downstream NLP tasks. However, most PLMs are trained on clean texts and are compromised inevitably to ASR errors. As shown in Figure 1, as WER gets higher, the SLU model suffers severe classification performance degradation (from 95.69% to 56.71%). Taking ASR hypotheses as inputs may introduce an issue that words in an utterance may be misrecognized. For instance, *fair* and *fare* are acoustically similar, so an ASR system may fail to distinguish between them, resulting in a substitution error. Such substitution errors might be recovered by humans because humans are aware of the acoustic confusability of words. However, the errors may significantly degrade the testing performance when the models are trained on oracle transcripts (*i.e.*, clean texts). In order to enhance the ASR robustness in contextualized word embeddings, Chang and Chen (2022) first introduced contrastive pre-training objectives with only textual information as input. However, they neglect finer-level alignment between the oracle transcript and the ASR hypothesis.

In this work, we propose a novel two-stage multi-grained contrastive learning framework (MCLF) for ASR-robust SLU (illustrated in Figure 2). Specifically, in the first stage, we devise a fine-grained interactive module to align the tokens of the oracle transcript with the ones of the ASR hypothesis. We also leverage contrastive learning to facilitate global representation learning at the utterance level. Inspired by Fang et al. (2020b) that consider the phonetic confusion, we explore various task-specific *data augmentation (DA)* methods to mitigate the deficiency of large amounts of labeled data for the PLM training procedure.[1] In the second stage, we fine-tune the pre-trained MCLF model with several robust training techniques. We conduct extensive experiments on four benchmark datasets (*i.e.*, SLURP, ATIS, TREC6, and TREC50) and four BERT-like backbone models. Experimen-

tal results and detailed analyses demonstrate the superiority and competitiveness of MCLF.

In summary, the contributions of this paper are three-fold: 1) We propose MCLF, which is the first attempt to combine global contrastive and fine-grained contrastive objectives for establishing ASR-robust SLU models; 2) Experiments on four backbone models and four datasets demonstrate the effectiveness of our proposed method; 3) Detailed analysis shows that our proposed framework brings effectiveness to the robustness of noise data.

## 2 Approach

In this section, we first detailedly elaborate on the pre-training tasks used in MCLF (Figure 3) in Section 2.1. Then we introduce the proposed data augmentation methods for contrastive learning in the pre-training stage (Section 2.2). Finally, we briefly describe the fine-tuning stage in Section 2.3.

### 2.1 Pre-training for MCLF

In Figure 3a, we illustrate the architecture of MCLF as well as the three pre-training objectives, namely masked language modeling ($\mathcal{L}_{mlm}$), global contrastive learning ($\mathcal{L}_g$), and fine-grained contrastive learning ($\mathcal{L}_f$). The combination of $\mathcal{L}_g$ and $\mathcal{L}_f$ is called multi-grained contrastive learning.

#### 2.1.1 Global Contrastive Learning

Contrastive learning has been proven effective to learn better representations in both computer vision (Chen et al., 2020; Li et al., 2021; Yao et al., 2022) and natural language processing (Gao et al., 2021; Kim et al., 2021b) research. It aims to learn highly discriminative features by pulling semantically related samples together and pushing apart unrelated ones. Here we adopt contrastive learning to learn the ASR-robust utterance encoder $f$, which is initialized with the original RoBERTa$_{\text{base}}$ model.

Assume a training batch with $B$ paired examples $\mathcal{D} = \{(x_i^o, x_i^a)\}_{i=1}^{B}$, where $(x_i^o, x_i^a)$ denotes the $i$-th clean oracle transcript and ASR hypothesis pair. They are both fed into the shared encoder, and $\mathbf{h}_i^o$ and $\mathbf{h}_i^a$ represent the $\ell_2$ normalized output embedding of the [CLS] token, respectively. We then apply contrastive learning over $\mathbf{h}_i^o$ and $\mathbf{h}_i^a$ by optimizing a symmetric InfoNCE loss. The ASR-to-Oracle contrastive loss $\mathcal{L}_{a2o}$ is:

$$\mathcal{L}_{a2o} = -\frac{1}{B} \sum_{i=1}^{B} \log \frac{e^{(\mathbf{h}_i^{o\top} \mathbf{h}_i^a / \tau_c)}}{\sum_j e^{(\mathbf{h}_i^{o\top} \mathbf{h}_j^a / \tau_c)}},$$

---

[1]More DA details are in Section 2.2 and Appendix B.

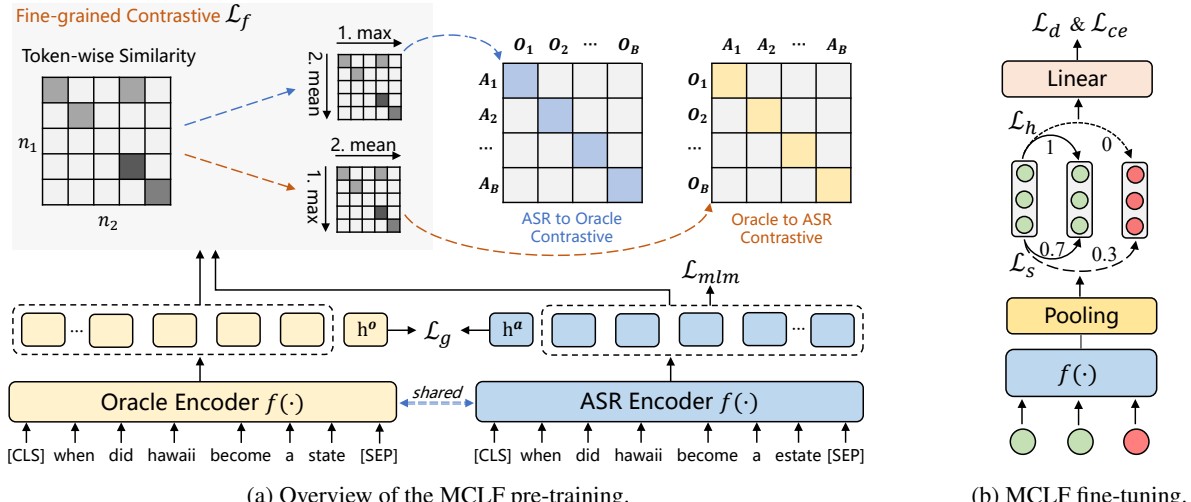

(a) Overview of the MCLF pre-training.      (b) MCLF fine-tuning.

Figure 3: (a) Overall schematic diagram of MCLF pre-training. MCLF is a dual-stream model consisting of a single weight-shared encoder $f$. Besides the canonical objective masked language modeling ($\mathcal{L}_{mlm}$), we introduce ASR-Oracle global contrastive loss ($\mathcal{L}_g$) and a novel fine-grained contrastive loss ($\mathcal{L}_f$) to align the representations of a clean-noisy utterance pair at both instance-level and token-level. (b) The fine-tuning objective includes a cross-entropy loss ($\mathcal{L}_{ce}$), two supervised contrastive learning loss ($\mathcal{L}_s$ and $\mathcal{L}_h$), and a self-distillation loss ($\mathcal{L}_d$).

where $\tau_c = 0.2$ is a temperature parameter. Similarly, the Oracle-to-ASR contrastive loss $\mathcal{L}_{o2a}$ is:

$$\mathcal{L}_{o2a} = -\frac{1}{B} \sum_{i=1}^{B} \log \frac{e^{(\mathbf{h}_i^{o\top} \mathbf{h}_i^a / \tau_c)}}{\sum_j e^{(\mathbf{h}_j^{o\top} \mathbf{h}_i^a / \tau_c)}},$$

thus the ASR-Oracle global contrastive loss is:

$$\mathcal{L}_g = \frac{1}{2}(\mathcal{L}_{a2o} + \mathcal{L}_{o2a}). \tag{1}$$

### 2.1.2 Fine-grained Contrastive Learning

Besides the global contrastive alignment, it remains a challenge to establish finer-grained token-level ASR-Oracle alignment. Recent work CASLU (Wang et al., 2022b) proposes to utilize a single cross-attention layer to explicitly model the fine-grained interactions between ASR hypotheses and phonemes sequence. However, it still relies on additional phoneme information and is inefficient. To this end, inspired by FILIP (Yao et al., 2022) that learns fine-grained image-text alignment through a cross-modal late interaction mechanism, we design the *fine-grained contrastive learning* which strengthens the model's robustness towards the local word error caused by phonetic confusion.

Specially, denote $n_1$ and $n_2$ as the token number of the $i$-th ASR hypothesis and $j$-th Oracle transcript, respectively, and the corresponding encoded features are $f(x_i^a) \in \mathbb{R}^{n_1 \times d}$ and $f(x_j^o) \in \mathbb{R}^{n_2 \times d}$. For the $k$-th ASR token, we compute its similarity with all oracle tokens of $x_j^o$, and use the largest one

$m_k^a = \max_{0 \le r < n_2} [f(x_i^a)]_k^\top [f(x_j^o)]_r$ as its token-wise maximum similarity with $x_j^o$. We then use the average token-wise maximum similarity of all non-padded tokens in the ASR hypothesis (resp. Oracle transcript) as the similarity of ASR-to-Oracle (resp. Oracle-to-ASR). The similarity of the $i$-th ASR hypothesis to the $j$-th Oracle transcript can thus be formulated as:

$$s_{i,j}^a = \frac{1}{n_1} \sum_{k=1}^{n_1} [f(x_i^a)]_k^\top [f(x_j^o)]_{m_k^a}. \tag{2}$$

Similarly, the similarity of the $j$-th Oracle transcript to the $i$-th ASR hypothesis is

$$s_{i,j}^o = \frac{1}{n_2} \sum_{k=1}^{n_2} [f(x_i^a)]_{m_k^o}^\top [f(x_j^o)]_k, \tag{3}$$

where $m_k^o = \arg\max_{0 \le r < n_1} [f(x_i^a)]_r^\top [f(x_j^o)]_k$. Note that $s_{i,j}^a$ in Equation (2) does not necessarily equal $s_{i,j}^o$ in Equation (3). Thus the fine-grained contrastive loss is:

$$\mathcal{L}_f = -\frac{1}{2B} \sum_{i=1}^{B} [\log \frac{e^{(s_{i,i}^a / \tau_c)}}{\sum_j e^{(s_{i,j}^a / \tau_c)}} + \\ \log \frac{e^{(s_{i,i}^o / \tau_c)}}{\sum_j e^{(s_{j,i}^o / \tau_c)}}]. \tag{4}$$

### 2.1.3 Overall Pre-training Objective

Following Gururangan et al. (2020) that leverages the masked language modeling (MLM) loss

for Domain-Adaptive Pretraining on downstream datasets, we keep the MLM loss in the pre-training for MCLF. Specially, the overall pre-training loss function is formulated as:

$$\mathcal{L}_p = \mathcal{L}_g + \lambda_f \mathcal{L}_f + \lambda_m \mathcal{L}_{mlm}, \qquad (5)$$

where the $\lambda_f$ and $\lambda_m$ are weight parameters.

## 2.2 Data Augmentation for Contrastive Learning in Pre-training

Traditionally, ASR system uses Word Error Rate (WER) as the performance metric to measure the percentage of incorrectly transcribed words. It is defined as follows:

$$\text{WER} = \frac{I + D + S}{N},$$

where $I$, $D$, and $S$ denote the number of inserted words, deleted words, and substitution words in the ASR hypothesis, respectively. $N$ is the number of words in the oracle text. From the perspective of WER, ASR errors can be divided into three categories (Fang et al., 2020b): insertions ($I$), deletions ($D$), and substitutions ($S$).

In this work, we design different data augmentation (DA) methods for contrastive pre-training following different ASR error types. 1) **Insertion DA.** According to preliminary experiments, we find that some words are easier to appear in the ASR hypothesis. So we augment the data by repeating the common words selected from the oracle transcript of training set. 2) **Deletion DA.** We randomly delete some words from the oracle transcript. 3) **Substitution DA.** We randomly replace some words from the oracle transcript.

First, we obtain the insertion words set and substitution words set (*i.e.*, confusion set) which acquires a new set of words that each word is most likely to be replaced with based on the training set. The construction procedure for insertion set and confusion set is in Algorithm 1. And we sort all of the confusion list for each word in descending order of longest common subsequence. Detailed sorting algorithm can be seen in Algorithm 2 in Appendix. Then, we perform different DA methods (i.e., Insertion/Deletion/Substitution) on part of sentences position. The target of position selection is to let WER reach a certain value. Empirically, for sentences with length less than 5, we keep it unchanged, for sentences with length between 5 and 10, we perform DA in one random position, for sentences with length larger than 10, we perform DA in two random positions.

---

**Algorithm 1:** Procedure for Insertion Set and Confusion Set Construction

**Input**: Training set $\mathcal{D} = \{(O_t, A_t)\}_{t=1}^{T}$
**Params**: $O_t$: the $t^{th}$ Oracle transcript
  $A_t$: the $t^{th}$ ASR hypothesis
  $T$: the size of training set
**Output**: $I$: the Insertion set
  $C$: the Confusion set
1: $I \leftarrow \varnothing; C \leftarrow \varnothing$
2: **for** $t \leftarrow 1$ to $T$ **do**
3:    $I_t \leftarrow \varnothing$
4:    $m \leftarrow length(O_t); n \leftarrow length(A_t)$
5:    **for** $i \leftarrow 1$ to $n$ **do**
6:      **if** $[A_t]_i \notin O_t$ **then**
7:        $I_t.\text{Insert}([A_t]_i)$
8:      **end if**
9:    **end for**
10:   $I \leftarrow I \cup I_t$
11:   **for** $i \leftarrow 1$ to $m$ **do**
12:     **if** $[O_t]_i \notin A_t$ **then**
13:       $C[[O_t]_i] \leftarrow C[[O_t]_i] \cup I_t$
14:     **end if**
15:   **end for**
16: **end for**
17: *// Sort each word list in descending order of longest common subsequence (LCS)*
18: Keys $\leftarrow$ C.keys()
19: **for** key in Keys **do**
20:   C[key].sort(LCS(C[key], key))
21: **end for**
22: **return** $I, C$

---

## 2.3 Fine-tuning MCLF for SLU Task

We further adapt the pre-trained MCLF model to the corresponding SLU datasets in fine-tuning.

Following Gunel et al. (2021), we leverage the supervised contrastive learning (SCL) objective, which is meant to pull the examples of the same label closer and push away the examples from other labels. As shown in Figure 3b, two examples with green color (the same color represents the same label) should be closer in the latent space, and be pulled away from the red example. Such technique can make full use of the hard label information during the training process. Denote $\mathbf{h}_i$ and $\mathbf{h}_j$ as the Mean-Max pooled features (Zhang et al., 2018) of all the output tokens from the utterance encoder.

The hard SCL loss is:

$$\mathcal{L}_h = -\frac{1}{N} \sum_i^N \sum_{i \neq j}^N \mathbf{1}_{y_i = y_j} \log \frac{e^{(\mathbf{h}_i^\top \mathbf{h}_j / \tau_c)}}{\sum_{k \neq i}^N e^{(\mathbf{h}_i^\top \mathbf{h}_k / \tau_c)}}.$$

Moreover, we adopt Chang and Chen (2022) that utilizes self-distillation to mitigate the impact of label noises in the training set by minimizing the KL divergence between current step prediction and the previous one. Denote $p_i^t = P(y_i|x_i, t)$ as the posterior probability of data $x_i$ predicted by the model followed by a single linear layer at the $t$-th epoch. The loss is formulated as:

$$\mathcal{L}_d = \frac{1}{N} \sum_i^N \mathrm{KL}_{\tau_d}(p_i^{t-1} || p_i^t),$$

where $p_i^0$ is the one-hot vector of the label $y_i$.

We also consider using the soft supervised contrastive learning as a complement of $\mathcal{L}_h$ to further relieve the label noise (Yun et al., 2020):

$$\mathcal{L}_s = -\frac{1}{N} \sum_i^N \sum_{i \neq j}^N p_i^{t-1} p_j^{t-1} \log \frac{e^{(\mathbf{h}_i^\top \mathbf{h}_j / \tau_c)}}{\sum_{k \neq i}^N e^{(\mathbf{h}_i^\top \mathbf{h}_k / \tau_c)}}.$$

Finally, the overall fine-tuning loss function is:

$$\mathcal{L}_{ft} = \mathcal{L}_{ce} + \lambda_{sc}(\mathcal{L}_h + \lambda_d \mathcal{L}_s) + \lambda_d \mathcal{L}_d, \quad (6)$$

where $\mathcal{L}_{ce}$ is the cross-entropy loss, $\lambda_{sc}$ and $\lambda_d$ are weight parameters.

# 3 Experiments

In this section, we evaluate the effectiveness of the proposed method on four benchmark datasets, *i.e.*, **SLURP** (Bastianelli et al., 2020), **ATIS** (Hemphill et al., 1990), **TREC6** and **TREC50** (Li and Roth, 2002). Data statistics are shown in Table 1.

| Dataset | #Class | Avg. Length | Train | Test | WER |
|---------|--------|-------------|-------|------|-----|
| **SLURP** | $18 \times 46$ | 6.93 | 50,628 | 10,992 | 25% |
| **ATIS** | 22 | 11.14 | 4,978 | 893 | 29.11% |
| **TREC6** | 6 | 8.89 | 5,452 | 500 | 32.93% |
| **TREC50** | 50 | 8.89 | 5,452 | 500 | 32.93% |

Table 1: Datasets statistics information.

## 3.1 Datasets

**SLURP** SLURP is a publicly available multi-domain SLU dataset with a collection of $\sim$72k audio recordings. It includes 18 different scenarios and 46 defined actions. Joint accuracy of both scenario and action is used as the evaluation metric of SLURP. Following Chang and Chen (2022), we use Google Web API (an off-the-shelf ASR system) to obtain the ASR hypothesis from the audio and adopt the same dataset split way. The median WER is 25% for Google Web API.

**ATIS** ATIS is a widely used SLU dataset with 22 different intents on flight reservation. The average WER for ATIS is 29.11%.

**TREC6 / TREC50** TREC is a question classification dataset with two versions, *i.e.*, TREC6 (6 classes) and TREC50 (50 classes). The difference is that the class labels of TREC50 are more fine-grained than TREC6. The average WER for both datasets is 32.93%. The ASR hypotheses of both ATIS and TREC are synthesized by TTS and later transcribed by the ASR system, and they have been released by PhonemeBERT[2].

## 3.2 Experimental Settings

**Backbone Models** In our experiments, we choose four pre-trained language models with different architectures and pre-trained tasks as backbones, *i.e.*, 1) RoBERTa$_{\text{base}}$ (Liu et al., 2019b); 2) BERT$_{\text{base}}$ (Devlin et al., 2019); 3) ELECTRA$_{\text{small}}$ (Clark et al., 2020); 4) TinyBERT$_4$ (Jiao et al., 2020). BERT is the most widely used PLM in various NLP tasks. RoBERTa improves the pre-training of BERT with various optimization techniques. ELECTRA replaces the MLM task in BERT with the more compute-efficient replaced token detection pre-training task. TinyBERT accelerates inference and reduces the model size compared with BERT through a novel Transformer distillation method. To demonstrate the consistent improvement brought by the proposed multi-grained contrastive objective, we conduct experiments on all different backbones while keeping BERT's related training parameters fixed, as can be seen in Appendix A.

**Training Details** We follow Chang and Chen (2022) for the data split and preprocessing. The training batch size is selected from {32, 64, 128} for each dataset. For each experimental setting, we pre-train the model with 20000 steps and then fine-tune it on the corresponding ASR hypotheses with 10 epochs. Unless otherwise specified, all

---

[2]https://github.com/
Observeai-Research/Phoneme-BERT

| PLM | Global CL | Fine-grained CL | SLURP | ATIS | TREC6 | TREC50 | Average |
|---|---|---|---|---|---|---|---|
| RoBERTa$_{base}$ | ✗ | ✗ | $84.04_{\pm0.19}$ | $94.42_{\pm0.54}$ | $84.76_{\pm0.67}$ | $75.08_{\pm0.27}$ | 84.58 |
| | ✓ | ✗ | $85.33_{\pm0.18}$ | $94.93_{\pm0.46}$ | $87.04_{\pm0.53}$ | $76.20_{\pm0.32}$ | 85.88 |
| | ✗ | ✓ | $85.01_{\pm0.14}$ | $94.58_{\pm0.50}$ | $85.76_{\pm0.37}$ | $78.72_{\pm0.78}$ | 86.02 |
| | ✓ | ✓ | $\mathbf{85.39}_{\pm0.28}$ | $\mathbf{95.22}_{\pm0.55}$ | $\mathbf{87.00}_{\pm0.52}$ | $\mathbf{78.84}_{\pm1.11}$ | **86.62** |
| BERT$_{base}$ | ✗ | ✗ | $83.87_{\pm0.24}$ | $94.27_{\pm0.34}$ | $85.40_{\pm0.28}$ | $74.56_{\pm0.45}$ | 84.53 |
| | ✓ | ✗ | $84.44_{\pm0.16}$ | $95.15_{\pm0.33}$ | $86.12_{\pm0.37}$ | $76.47_{\pm0.98}$ | 85.55 |
| | ✗ | ✓ | $84.13_{\pm0.14}$ | $94.83_{\pm0.44}$ | $86.04_{\pm0.62}$ | $78.40_{\pm0.68}$ | 85.85 |
| | ✓ | ✓ | $\mathbf{84.85}_{\pm0.31}$ | $94.95_{\pm0.20}$ | $\mathbf{86.40}_{\pm0.85}$ | $\mathbf{78.92}_{\pm0.20}$ | **86.28** |
| ELECTRA$_{small}$ | ✗ | ✗ | $69.77_{\pm0.60}$ | $92.67_{\pm0.56}$ | $82.44_{\pm0.64}$ | $58.32_{\pm0.75}$ | 75.80 |
| | ✓ | ✗ | $74.82_{\pm0.40}$ | $92.83_{\pm0.84}$ | $85.08_{\pm0.91}$ | $61.12_{\pm0.74}$ | 78.46 |
| | ✗ | ✓ | $82.70_{\pm0.17}$ | $94.25_{\pm0.26}$ | $84.20_{\pm0.74}$ | $72.30_{\pm1.26}$ | 83.36 |
| | ✓ | ✓ | $\mathbf{82.76}_{\pm0.24}$ | $\mathbf{95.00}_{\pm0.24}$ | $\mathbf{85.56}_{\pm1.09}$ | $\mathbf{74.52}_{\pm0.30}$ | **84.46** |
| TinyBERT$_4$ | ✗ | ✗ | $72.05_{\pm0.56}$ | $88.45_{\pm0.76}$ | $77.84_{\pm0.46}$ | $55.28_{\pm0.37}$ | 73.41 |
| | ✓ | ✗ | $78.14_{\pm0.13}$ | $93.46_{\pm0.15}$ | $82.90_{\pm0.30}$ | $64.10_{\pm0.30}$ | 79.65 |
| | ✗ | ✓ | $81.80_{\pm0.12}$ | $95.00_{\pm0.26}$ | $82.64_{\pm0.81}$ | $70.60_{\pm0.66}$ | 82.51 |
| | ✓ | ✓ | $\mathbf{81.83}_{\pm0.15}$ | $\mathbf{95.35}_{\pm0.24}$ | $\mathbf{84.04}_{\pm0.96}$ | $\mathbf{72.60}_{\pm0.85}$ | **83.46** |

Table 2: Average accuracy (%) across 5 seeds on the benchmarks under different pre-trained language models and their standard deviation. For each PLM, results in row 1 are with fine-tuning only, and the results of row 2 to 4 are pre-trained first (all equipped with MLM loss by default) and then fine-tuned.

experimental training processes are kept consistent. Detailed experimental settings are in Appendix A.

| Method | SLURP | ATIS | TREC6 | TREC50 |
|---|---|---|---|---|
| RoBERTa | 83.97 | 94.53 | 84.08 | 75.02 |
| PhonemeBERT | 83.78 | 94.83 | 85.96 | 76.16 |
| SimCSE | 84.47 | 94.07 | 84.92 | 74.82 |
| SpokenCSE | 85.26 | 95.10 | 86.36 | 76.20 |
| **MCLF (Ours)** | **85.39** | **95.22** | **87.00** | **78.84** |

Table 3: Accuracy (%) comparison of baseline works and the proposed MCLF on four datasets. For a fair comparison, all of the methods in this table employ RoBERTa$_{base}$ model as the backbone.

## 3.3 Main Results

In Table 2, we show the performance of four backbones introduced in Section 3.2 under different pre-training objective settings in Equation (5), *i.e.*, 1) no pre-training stage; 2) with only $\mathcal{L}_g$ and $\mathcal{L}_{mlm}$; 3) with only $\mathcal{L}_f$ and $\mathcal{L}_{mlm}$; and 4) with the full pre-training objective. And they correspond to row 1 $\sim$ row 4 for each backbone, respectively. As can be seen, when compared with the baseline result in row 1, using global contrastive learning (global CL) and fine-grained CL respectively in pre-training can boost the performance of all datasets for all backbone models, and combining both $\mathcal{L}_g$ and $\mathcal{L}_f$ at the same time brings further gains, which indicates the effectiveness of the proposed multi-grained contrastive learning. Another observation is that using our proposed MCLF upon a smaller backbone (*i.e.*, parameters scale-wise) usually has a higher accuracy gain than upon the larger one, this indicates that MCLF is more efficient to dig the potential of smaller models and has the potential to make up for the inefficient models' capacity. Fine-grained CL, most of the time, performs better than those that only employ global CL under smaller backbones.

## 3.4 Comparison with Baselines

To validate the superiority of MCLF, in this section, we implement various previous methods as baselines for comparison:

- RoBERTa (Liu et al., 2019b): A RoBERTa$_{base}$ model directly fine-tuned on the SLU tasks using cross-entropy loss, and with ASR hypothesis as input.

- PhonemeBERT (Sundararaman et al., 2021): A RoBERTa$_{base}$ model first jointly trained on ASR hypothesis augmented with phoneme sequence (generated from the ASR hypothesis with Phonemizer tool[3]) and then fine-tuned for SLU tasks with cross-entropy loss.

---

[3] https://github.com/bootphon/phonemizer

- SimCSE (Gao et al., 2021): We first simply feed the same ASR hypothesis *twice* and get the two embeddings with different dropout noise (both $p = 0.1$) for computing $\mathcal{L}_g$ in Equation (1), then it was fine-tuned on the same dataset with cross-entropy loss.

- SpokenCSE (Chang and Chen, 2022): The closest method to our MCLF. A major difference is that they only consider global contrastive learning during pre-training while ignoring finer-grained interactions (*e.g.*, token-level ASR-Oracle alignment). And we introduce the fine-grained contrastive objective $\mathcal{L}_f$ in Equation (4) to alleviate this problem.

As shown in Table 3, MCLF surpasses previous methods on all four datasets. Among them, PhonemeBERT, SimCSE, and SpokenCSE focus on feature representation in pre-training. Phoneme-BERT needs extra phoneme sequence as input and is less effective on SLURP. SpokenCSE leverages global contrastive learning by taking oracle transcripts and the corresponding ASR hypotheses as input and gets rid of the dependence on phoneme information. By introducing an extra novel fine-grained contrastive objective together with the DA method, our MCLF improves considerably over the latest SpokenCSE as well as all other baselines.

## 4 Discussions

### 4.1 UMAP Visualization

To further understand how the proposed method plays an important role, we extract the 768-*d* utterance embeddings from the TREC6 dataset and use the Uniform Manifold Approximation and Projection (UMAP) (McInnes and Healy, 2018) algorithm to project these embeddings onto a 2-*d* plane in Figure 4. First, from Figure 4a and Figure 4b, we can see that the semantics of ASR utterances are very scattered, farther than oracle utterances. Second, Figure 4a and 4c show the representations of ASR hypothesis on RoBERTa$_{base}$ and MCLF, respectively. We can see that most of the ASR utterances with same class are pulled together in both models. In RoBERTa$_{base}$, there are some ASR utterances that cannot be assigned to specific classes, while nearly all the utterances in MCLF model are pulled closer. Third, from Figure 4c and Figure 4d, even in MCLF, there are still some ASR utterances that are indistinguishable, which we believe that

it is due to the ASR errors that make them indistinguishable, for example, we found two oracle utterances with different labels but they share same ASR assumption in the dataset.

### 4.2 Effect of Pooling Strategy in Fine-tuning

In Table 4, we evaluate the effect of different feature pooling strategies in fine-tuning. Specifically, we first pre-train the BERT$_{base}$ model with several contrastive learning objectives, then during fine-tuning, we try different output features pooling strategies for utterance representations, *i.e.*, 1) `[CLS]`: global `[CLS]` representation; 2) `Mean`: averaging of all output token representations; 3) `Mean-Max`: concatenating the mean and maximum of all token features (Zhang et al., 2018).

As can be seen, compared to the global `[CLS]` representation, method `Mean` can further improve performance. Especially, the model got the best results using `Mean-Max` pooling. Thus, we can conclude that, after adding fine-grained contrastive learning loss, layer `Mean` pooling strategy outperforms `[CLS]`. Intuitively, we attribute this to that the fine-grained interaction provides fruitful information to the tokens. This proves that the whole sentence representation is more meaningful and distinguishable with fine-grained contrastive learning.

### 4.3 Case Study on Different Error Types

We take several typical ASR errors from TREC6 for analysis in Table 5. The predictions are collected based on the RoBERTa$_{base}$ and MCLF predictions. For six labels of TREC6, the total mismatch predictions (*e.g.*, the golden label is **HUM** but predicted as **LOC**) can be up to 30. Here, we list examples for analyzing the origin of the ASR errors between and after employing the MCLF. Take the first Oracle-ASR pair for example, when the key information *Philippines* is deleted, the baseline model tends to treat the whole utterance as **DESC** label, while the MCLF can capture fine-grained information like word *in* thus to predict true label as **LOC**. There are many such examples. However, we also find some interesting cases that are hard to solve even employing the MCLF, for example, in the third example, both the RoBERTa$_{base}$ and MCLF mispredict the label as **HUM**, because the ASR hypothesis is *what is nikki*, for this example, the ASR error will cause both the local and global information of the utterance to change. In such a case, even MCLF will not have much effect.

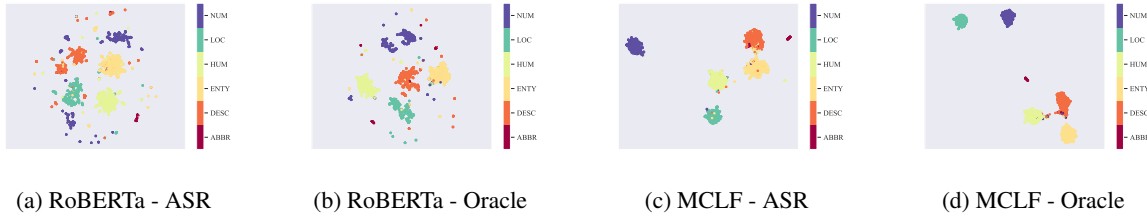

| (a) RoBERTa - ASR | (b) RoBERTa - Oracle | (c) MCLF - ASR | (d) MCLF - Oracle |

Figure 4: UMAP visualization of ASR and Oracle utterances on RoBERTa and MCLF model on TREC6 dataset.

| BERT$_{base}$ | SLURP | | | TREC6 | | |
|---|---|---|---|---|---|---|
| | [CLS] | Mean | Mean-Max | [CLS] | Mean | Mean-Max |
| w/ Global CL | 83.98 | 84.10 | **84.44** | 85.91 | 85.95 | **86.12** |
| w/ Fine-grained CL | 83.71 | 83.93 | **84.13** | 85.68 | 85.90 | **86.04** |
| w/ Multi-grained CL | 84.39 | 84.71 | **84.85** | 85.80 | 86.22 | **86.40** |

Table 4: Performance on SLURP and TREC6 datasets under different output feature pooling strategies.

| Example | ASR Error Type | Golden Label | RoBERTa Prediction | MCLF Prediction |
|---|---|---|---|---|
| O: What hemisphere is the Philippines in
A: What hemisphere is the in | Deletion | LOC | DESC | LOC |
| O: Why is a ladybug helpful
A: Why is a lady bug helpful | Insertion | DESC | ENTY | DESC |
| O: What is nicotine
A: What is nikki | Substitution | DESC | HUM | HUM |

Table 5: ASR error analysis on TREC6 test dataset. "O" and "A" denote the Oracle transcript and ASR hypothesis, respectively. Label in red denotes an error while the one in blue denotes the correct.

## 5 Related Work

SLU is an essential task for machines to infer correct semantic meaning (*e.g.*, intent detection, slot filling) from human speech (Huang et al., 2020; Zhou et al., 2020; Huang and Chen, 2020; Huang et al., 2022b; Chen et al., 2022b; Cheng et al., 2023b,d,e; Zhu et al., 2023a). Traditionally, it can be solved by fine-tuning NLP models (especially PLMs) with the ASR hypothesis as input (Wang et al., 2022a). However, the ASR hypothesis often contains errors caused by ASR systems. If the SLU models are trained with clean texts, the performance will be poor when predicting using noisy text as input. Also, models trained with noisy texts may suffer performance degradation over datasets without ASR errors.

The core problem lies in improving the ASR robustness for SLU models, and several types of methods have been proposed to address this issue. One stream treats texts with or without noise as two modalities. For example, Tan and Ling (2019) proposed to rebuild tuples between ASR results and

correct candidates by leveraging the minimum edit distance (MED). Ruan et al. (2020) introduced a novel loss to minimize the distribution differences between correct and incorrect output texts. SpokenCSE (Chang and Chen, 2022) aimed to learn the invariant representations between clean text and erroneous hypothesis by utilizing a contrastive objective to adapt PLM to ASR results. Another stream considers making full use of the ASR output information (*e.g.*, N-best hypothesis, phoneme sequence). For example, Zhu et al. (2021) proposed two novel approaches (*i.e.*, N-best Plain, N-best Alignment) that combine the N-best hypothesis of ASR as input for error correction. Wang et al. (2022b) proposed to trigger the fine-grained interactions between phoneme and word embeddings to derive phonetic-aware ASR-robust features. Cheng et al. (2023a) used manual transcripts to improve ASR robustness by mutual learning. Xie et al. (2023) proposed to incorporate syntax information into SLU system to strengthen ASR-robust ability by constraining attention scopes based on relationships within the syntactic structure. Cheng et al.

(2023c) utilized cross attention to fuse the features between manual transcripts and ASR transcripts and contrastive attention to capture unique features of ASR transcripts.

Considering that the ASR output information may not be easily obtained due to the constraint of ASR systems, we adopt the first stream in this paper. In addition to following Chang and Chen (2022) that builds alignment between ASR hypotheses and the oracle transcripts through a contrastive learning objective, we design the fine-grained contrastive learning which strengthens the model's robustness towards the local word error caused by phonetic confusion, and it serves as a complement to the global contrastive learning. The combination of both types of contrastive objectives is called multi-grained contrastive learning, aiming to learn both utterance and token level alignment between ASR hypotheses and the oracle transcripts, which mines fruitful utterance-level and token-level information from downstream datasets.

## 6 Conclusion

In this paper, we propose MCLF, an ASR-robust SLU model towards multi-grained Oracle-ASR contrastive pre-training. Besides, several data augmentation techniques are used to facilitate the contrastive pre-training stage. To our best knowledge, this is the first attempt to improve the ASR robustness of SLU models by simultaneously exploiting both global and fine-grained features alignment between ASR hypothesis and oracle transcripts. Empirical results on four benchmark datasets show the effectiveness of the proposed MCLF and the superiority compared with several baseline works.

## Limitations

As with any research, MCLF is not without limitations. We split the discussion into limitations that are inherent to our method, and limitations of our present study, the latter can be overcome by extensions of our work. Previous studies (*e.g.*, SpokenCSE (Chang and Chen, 2022)) conducted contrastive learning with ASR and oracle utterance pairs using the Transformer-based model. Our empirical results can be seen as further mining the potential of contrastive learning and illustrating the generalization of our method under multiple baseline backbones. Another technical limitation is that our method requires the cost of additional training data, and while we have shown that our

experiments can be performed on a single NVIDIA V100 GPU, it still incurs additional training time and cost. However, this does not change the range of devices on which the technique can be applied, since data augmentation does not change the size of the entire SLU model, and at the same time, the inference speed is basically unchanged.

## Ethics Statement

Our goal in developing an ASR-robust SLU model is to enable practical intelligent assistant applications more useful and appealing to human users through robust semantic recognition with ASR erroneous outputs. For example, for people with different accents, different language habits, and whether they use slang, their inputs to the ASR system will be very different. So it is difficult for the ASR model to fully consider all of these situations and correctly identify all spoken errors. Therefore, the design of a robust spoken language understanding system for ASR is very important. And our proposed approach based on multi-granularity contrastive learning effectively adapts the SLU model to the error of the ASR system. Overall, ASR error is a crucial problem in application scenarios such as intelligent speech assistant, and having an ASR-robust SLU model that can filter such errors would have many positive applications.

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

## A  Experimental Details

The detailed hyperparameters for training MCLF are shown in Table 6.

| Config | Pre-training | Fine-tuning |
|---|---|---|
| Batchsize | {32, 64, 128} | {32, 64, 128} |
| Learning rate (lr) | 2e-5 | 2e-5 |
| Lr shedule | linear decay | linear decay |
| Gradient clipping | 1 | 1 |
| Dropout rate | 0.1 | 0.1 |
| Attention dropout | 0.1 | 0.1 |
| MLM ratio | 15% | - |
| $\lambda_f, \lambda_m$ | 0.1, 1.0 | - |
| $\lambda_{sc}, \lambda_d$ | - | 1.0, 10 |
| $\tau_c$ | 0.2 | 0.2 |
| $\tau_d$ | - | 5 |
| Data Augmentation | y | n |
| Training Steps/Epochs | 20000 Steps | 10 Epochs |

Table 6: Experimental setup for two stages in training four backbone models on four benchmark datasets.

## B  Data Augmentation Details

**LCS Sorting Algorihtm**  We show the Longest Common Subsequence (LCS) sorting details in Algorithm 2.

**Ablation Study**  To further illustrate the effect of our data augmentation approach, in this section, we show the ablation results of MCLF in Table 7 that remove the DA part, while keeping other training details unchanged. We find that DA is important as part of the training step for multi-grained contrastive learning objectives, and can help PLMs to learn more representative features.

| Ablation Settings | ATIS ($\Delta$Acc) |
|---|---|
| RoBERTa | 0 |
| G-CL: Insertion-DA | +0.2 |
| G-CL: Deletion-DA | +0.1 |
| G-CL: Substitution-DA | +0.2 |
| FG-CL: Insertion-DA | +0.2 |
| FG-CL: Deletion-DA | +0.3 |
| FG-CL: Substitution-DA | +0.3 |
| MCLF | +0.8 |

Table 7: Ablation study of DA on the MCLF.

## C  Effect of Training Batch Size for MCLF with Different Backbones

In this section, we analyze the effect of batch size in pre-train and fine-tune stages. We show the results on MCLF of different batch size of in Table 8.

---

**Algorithm 2:** LCS-based Sorting Algorithm for Utterance X and Y

**Input**: $X = <x_1, ...x_m>$
$\qquad\quad Y = <y_1, ...y_n>$
**Output**: $c[m, n]$: the LCS of $X$ and $Y$

```
1:  m ← length[X] ;  n ← length[Y]
2:  for i ←1 to m do
3:      c[i, 0] ← 0
4:  end for
5:  for j ←1 to n do
6:      c[0, j] ← 0
7:  end for
8:  for i ←1 to m do
9:      for j ←1 to n do
10:         if X[i] == Y[j] then
11:             c[i, j] ← c[i − 1, j − 1] + 1
12:         else
13:             c[i, j] ← max(c[i, j − 1], c[i − 1, j])
14:         end if
15:     end for
16: end for
17: return c[m, n]
```

We can see that, 1) different datasets do not share a uniform best batch size, while having different optimal batch sizes for optimal performance; 2) smaller dataset has larger results variance than larger one (*e.g.*, TREC to SLURP), showing that MCLF gets consistent performance when the training data is larger; 3) for ATIS, large fine-tune batch size get better results, for SLURP, large pre-trained batch size get better results.

## D  Implementation Details

We outline the key components of the fine-grained and global contrastive objectives in PyTorch. Please refer to Listing 1 for details.

| PLM | PT batch size | FT batch size | SLURP | ATIS | TREC6 | TREC50 |
|---|---|---|---|---|---|---|
| RoBERTa$_{base}$ | 32 | 32 | $85.14_{\pm0.16}$ | $94.32_{\pm0.36}$ | $86.40_{\pm0.83}$ | $77.56_{\pm0.94}$ |
| | 32 | 64 | $\mathbf{85.39}_{\pm0.28}$ | $94.95_{\pm0.40}$ | $86.76_{\pm0.99}$ | $77.88_{\pm0.47}$ |
| | 32 | 128 | $85.30_{\pm0.14}$ | $95.05_{\pm0.25}$ | $86.40_{\pm0.38}$ | $77.64_{\pm0.79}$ |
| | 64 | 32 | $85.33_{\pm0.27}$ | $94.12_{\pm0.38}$ | $86.76_{\pm0.64}$ | $77.20_{\pm0.72}$ |
| | 64 | 64 | $85.27_{\pm0.11}$ | $94.93_{\pm0.41}$ | $\mathbf{87.00}_{\pm0.52}$ | $78.12_{\pm0.60}$ |
| | 64 | 128 | $85.08_{\pm0.10}$ | $95.07_{\pm0.46}$ | $86.32_{\pm0.37}$ | $77.80_{\pm0.38}$ |
| | 128 | 32 | $85.24_{\pm0.15}$ | $94.67_{\pm0.94}$ | $86.52_{\pm0.73}$ | $77.92_{\pm1.08}$ |
| | 128 | 64 | $85.31_{\pm0.21}$ | $95.20_{\pm0.32}$ | $86.72_{\pm0.63}$ | $\mathbf{78.84}_{\pm1.11}$ |
| | 128 | 128 | $85.01_{\pm0.14}$ | $\mathbf{95.22}_{\pm0.55}$ | $86.20_{\pm0.63}$ | $78.28_{\pm0.83}$ |
| BERT$_{base}$ | 32 | 32 | $84.82_{\pm0.16}$ | $94.82_{\pm0.43}$ | $86.16_{\pm0.82}$ | $78.32_{\pm0.61}$ |
| | 32 | 64 | $84.54_{\pm0.18}$ | $94.75_{\pm0.35}$ | $85.88_{\pm0.41}$ | $78.20_{\pm0.61}$ |
| | 32 | 128 | $84.11_{\pm0.20}$ | $\mathbf{94.95}_{\pm0.20}$ | $85.96_{\pm0.71}$ | $78.04_{\pm0.54}$ |
| | 64 | 32 | $84.82_{\pm0.20}$ | $94.77_{\pm0.63}$ | $85.48_{\pm1.09}$ | $\mathbf{78.92}_{\pm0.20}$ |
| | 64 | 64 | $84.67_{\pm0.24}$ | $94.70_{\pm0.46}$ | $85.72_{\pm0.48}$ | $78.56_{\pm0.34}$ |
| | 64 | 128 | $84.20_{\pm0.20}$ | $\mathbf{94.95}_{\pm0.33}$ | $86.32_{\pm0.37}$ | $78.32_{\pm0.48}$ |
| | 128 | 32 | $\mathbf{84.85}_{\pm0.31}$ | $94.58_{\pm0.42}$ | $85.92_{\pm0.72}$ | $78.76_{\pm0.56}$ |
| | 128 | 64 | $84.81_{\pm0.19}$ | $94.10_{\pm0.48}$ | $\mathbf{86.40}_{\pm0.85}$ | $78.84_{\pm0.82}$ |
| | 128 | 128 | $84.11_{\pm0.15}$ | $94.80_{\pm0.39}$ | $86.24_{\pm0.93}$ | $\mathbf{78.92}_{\pm0.61}$ |
| ELECTRA$_{small}$ | 32 | 32 | $79.79_{\pm0.27}$ | $94.80_{\pm0.26}$ | $84.24_{\pm0.89}$ | $71.72_{\pm0.94}$ |
| | 32 | 64 | $81.48_{\pm0.21}$ | $94.55_{\pm0.43}$ | $84.44_{\pm0.53}$ | $72.76_{\pm1.11}$ |
| | 32 | 128 | $81.69_{\pm0.33}$ | $94.88_{\pm0.19}$ | $\mathbf{85.56}_{\pm1.09}$ | $73.52_{\pm0.65}$ |
| | 64 | 32 | $80.24_{\pm0.27}$ | $94.70_{\pm0.47}$ | $84.96_{\pm1.18}$ | $72.08_{\pm0.41}$ |
| | 64 | 64 | $81.81_{\pm0.11}$ | $94.82_{\pm0.30}$ | $84.80_{\pm1.51}$ | $74.24_{\pm0.39}$ |
| | 64 | 128 | $82.43_{\pm0.39}$ | $95.05_{\pm0.48}$ | $85.52_{\pm0.89}$ | $\mathbf{74.52}_{\pm0.30}$ |
| | 128 | 32 | $80.81_{\pm0.19}$ | $94.63_{\pm0.24}$ | $83.36_{\pm0.97}$ | $71.56_{\pm0.91}$ |
| | 128 | 64 | $82.19_{\pm0.18}$ | $94.45_{\pm0.19}$ | $84.64_{\pm0.86}$ | $73.56_{\pm0.66}$ |
| | 128 | 128 | $\mathbf{82.76}_{\pm0.24}$ | $\mathbf{95.00}_{\pm0.24}$ | $84.12_{\pm0.55}$ | $74.12_{\pm0.69}$ |
| TinyBERT$_4$ | 32 | 32 | $79.80_{\pm0.28}$ | $94.75_{\pm0.38}$ | $83.36_{\pm0.66}$ | $70.76_{\pm0.91}$ |
| | 32 | 64 | $81.55_{\pm0.22}$ | $94.35_{\pm0.32}$ | $83.48_{\pm0.70}$ | $71.40_{\pm0.72}$ |
| | 32 | 128 | $81.65_{\pm0.25}$ | $95.12_{\pm0.36}$ | $83.00_{\pm0.61}$ | $72.08_{\pm1.13}$ |
| | 64 | 32 | $80.15_{\pm0.22}$ | $94.55_{\pm0.50}$ | $83.68_{\pm1.00}$ | $71.36_{\pm0.75}$ |
| | 64 | 64 | $81.77_{\pm0.24}$ | $94.95_{\pm0.52}$ | $\mathbf{84.04}_{\pm0.96}$ | $72.16_{\pm0.89}$ |
| | 64 | 128 | $81.71_{\pm0.07}$ | $\mathbf{95.35}_{\pm0.24}$ | $83.32_{\pm0.57}$ | $\mathbf{72.60}_{\pm0.85}$ |
| | 128 | 32 | $80.42_{\pm0.19}$ | $94.75_{\pm0.38}$ | $83.68_{\pm0.57}$ | $70.12_{\pm0.32}$ |
| | 128 | 64 | $\mathbf{81.83}_{\pm0.21}$ | $95.30_{\pm0.47}$ | $83.84_{\pm0.45}$ | $70.84_{\pm0.32}$ |
| | 128 | 128 | $\mathbf{81.83}_{\pm0.15}$ | $95.22_{\pm0.19}$ | $83.32_{\pm0.30}$ | $71.20_{\pm0.67}$ |

Table 8: Results of four backbones on benchmark datasets under different pre-train and fine-tune batch sizes.

```python
import torch
import torch.nn as nn
import torch.nn.functional as F
import math

def token_wise_similarity(rep1, rep2, chunk_size=1024):
    batch_size1, n_token1, feat_dim = rep1.shape
    batch_size2, n_token2, _ = rep2.shape
    num_folds = math.ceil(batch_size2 / chunk_size)
    output = []
    for i in range(num_folds):
        rep2_seg = rep2[i * chunk_size:(i + 1) * chunk_size]
        out_i = rep1.reshape(-1, feat_dim) @ rep2_seg.reshape(-1, feat_dim).T
        out_i = out_i.reshape(batch_size1, n_token1, batch_size2, n_token2)
        out_i = out_i.max(3)[0].mean(1)
        output.append(out_i)
    return torch.cat(output, dim=1)

def get_2b_sim(asr, oracle):
    m1 = token_wise_similarity(asr, asr)
    m2 = token_wise_similarity(asr, oracle)
    m3 = token_wise_similarity(oracle, asr)
    m4 = token_wise_similarity(oracle, oracle)
    matrix_up = torch.cat([m1, m2], dim=1)
    matrix_down = torch.cat([m3, m4], dim=1)
    return torch.cat([matrix_up, matrix_down], dim=0)

class ContrastiveLoss(nn.Module):
    temperature = 0.05

    def l_ij(self, i, j, sim_mat, batch_size, z_i):
        numerator = torch.exp(sim_mat[i, j] / self.temperature)
        identity_except_i = torch.ones((2 * batch_size, )).scatter_(0, torch.tensor
    ([i]), 0.0).to(z_i.device)
        denominator = torch.sum(identity_except_i * torch.exp(sim_mat[i, :] / self.
    temperature))
        loss_ij = -torch.log(numerator / denominator)
        return loss_ij.squeeze(0)

    def forward(self, z_i, z_j, granularity):
        # z_j, z_j: global_shape=[b,d]; fine_shape=[b,n1/n2,d]
        batch_size = z_i.shape[0]
        if granularity == "global":
            rep = torch.cat([z_i, z_j], dim=0)
            sim_mat = F.cosine_similarity(rep.unsqueeze(1), rep.unsqueeze(0), dim=2)
        elif granularity == "fine":
            z_i = z_i / z_i.norm(dim=-1, keepdim=True)
            z_j = z_j / z_j.norm(dim=-1, keepdim=True)
            sim_mat = get_2b_sim(z_i, z_j)

        loss = 0.0
        for k in range(0, batch_size):
            loss = loss + self.l_ij(k, k + batch_size, sim_mat, batch_size, z_i)
            loss = loss + self.l_ij(k + batch_size, k, sim_mat, batch_size, z_i)
        return 1.0 / (2 * batch_size) * loss
```

Listing 1: Contrastive Loss Implementation