# OpenReview forum: "MCLF: A Multi-grained Contrastive Learning Framework for ASR-robust Spoken Language Understanding"
_EMNLP/2023/Conference — EMNLP 2023 Findings_

### Official Review · Reviewer_u9Ji · 2023-08-06

**Soundness:** 4

**Excitement:**

3: Ambivalent: It has merits (e.g., it reports state-of-the-art results, the idea is nice), but there are key weaknesses (e.g., it describes incremental work), and it can significantly benefit from another round of revision. However, I won't object to accepting it if my co-reviewers champion it.

**Missing References:**

References for E2E SLU models:
https://arxiv.org/abs/2111.14706
https://arxiv.org/abs/2008.06173

References for E2E SLU models incorporating pre-trained LM:
https://arxiv.org/abs/2207.06670
https://arxiv.org/abs/2010.13826

**Paper Topic And Main Contributions:**

The paper proposes enhancing the robustness of SLU systems to ASR errors. The authors utilize a two-stage multi-grained contrastive learning framework to adapt a pre-trained language model to ASR hypotheses of SLU datasets. They find that their approach helps improve performance on four publicly available SLU datasets.

**Reasons To Accept:**

1. Clear description of their methodology with mathematical formulations.
2. Good performance improvements, particularly when the PLM is small like TinyBERT

**Reasons To Reject:**

1. The paper lacks a comparison with a relevant line of research, namely E2E SLU systems (https://arxiv.org/abs/2111.14706, https://arxiv.org/abs/2008.06173), which avoid cascading errors from the ASR system. Even if there were no empirical comparisons, the authors should have provided some discussion on the advantages of their robustness approach over end-to-end approaches. Interestingly, E2E SLU approaches (https://arxiv.org/pdf/2111.14706.pdf) appear to perform better than their approach on the SLURP dataset, even without incorporating any pre-trained LM.
2. The main contribution of the paper is Fine-grained Contrastive Learning, as I understand it. It would have been insightful to include an error analysis to better understand the sources of performance improvements resulting from using fine-grained contrastive learning.

**Reproducibility:**

5: Could easily reproduce the results.

**Reviewer Confidence:**

5: Positive that my evaluation is correct. I read the paper very carefully and I am very familiar with related work.

---

> ### Author Rebuttal · Authors · 2023-08-29
>
> We extend our heartfelt appreciation for the thorough reviews you have provided. We value your insights and would like to offer detailed responses to your concerns.
>
>
> **On the concern about error analysis**
>
> - Thank you for your insightful feedback. We agree that providing an error analysis could offer a better understanding of the performance improvements emanating from the use of fine-grained contrastive learning. We value your comments and to address this, we have conducted three different ASR error analysis on the TREC6 test set. The table below presents a snapshot of this analysis.
>
> |Examples|ASR Error Type|Golden Label|RoBERTa Prediction |Global Contrastive Learning Prediction|MCLF Prediction|
> |:---------------------------|:---------------------|:--------------------|:---------------------|:-------------------|:---------------------|
> |O: "What hemisphere is the Philippines in" A: "What hemisphere is the in"|Deletion|LOC|DESC|LOC|LOC|
> |O: "Why is a ladybug helpful Insertion" A: "Why is a lady bug helpful|Insertion| DESC|ENTY|ENTY|DESC|
> |O: "What is nicotine" A: "What is nikki"|Substitution|DESC|HUM|HUM|HUM|
>
>
> - Take the first example to elaborate, the Oracle Transcript (O) "What hemisphere is the Philippines in?" becomes "What hemisphere is the in?" through ASR, due to the omission of "Philippines". This omission makes comprehension of the sentence meaning challenging for the traditional SLU backbone model (e.g., RoBERTa), leading to incorrect judgments.
>
> - We also discovered that the sentence vector distances in the mentioned example sentences are as such: Ex3 > Ex2 > Ex1 (measured using BERT-base-uncase [CLS] embedding). We noted that when using the original RoBERTa model, all three cases result in incorrect prediction; when using Global Contrastive Learning (CL), Ex1 is predicted correctly; and when applying MCLF, both Ex1 and Ex2 are accurately predicted. These demonstrations highlight the efficacy of the proposed MCLF model. In addition, it suggests that the farther the sentence embedding with ASR error is from the oracle sentence embedding, the less likely it is to be correctly predicted by CL-trained model, and the benefits of employing fine-grained CL are increasingly noticeable. We will include this analysis in the next version of the paper, along with a detailed discussion on the findings. We appreciate your suggestions and believe that such an analysis will indeed provide the readers with a richer perspective on the effectiveness of the proposed method.
>
>
>
> **On the concern about missing references**
>
> - We appreciate your insightful comments and the associated references. We agree that a discussion with end-to-end spoken language understanding (E2E SLU) systems would be valuable. Indeed, the E2E SLU systems can prevent cascading errors from the ASR system, potentially leading to better performance.
>
> - First, we would like to emphasize that our current focus primarily lies in the examination of performance robustness across different ASR environments. Our proposed method provides a greater degree of flexibility by decoupling the SLU components from the ASR modules, thus facilitating a more comprehensive analysis of ASR errors. Further, our methodology can be seamlessly integrated into a broad spectrum of existing ASR models and interfaces with less costs involved.
>
> - Upon close examination of the papers you recommended, we noted that the E2E SLU methods employed by [1] and [2] yielded impressive results on the FSC, SLURP, and SNIPS datasets. Though, it is worth highlighting that the fundamental differences between these approaches and ours. While E2E SLU designs aim to circumvent cascading errors from ASR systems, our strategy intentionally addresses these errors to enhance system robustness.
>
> - In conclusion, thanks for the value and significance of your suggestion. We will add discussion with references [1][2] in the next version and explore incorporating a broader range of comparisons and discussions related to E2E systems. Your constructive feedback has proved invaluable, and we extend our gratitude for drawing our attention to this important aspect.
>
> **References**
>
> [1] ESPNET-SLU: ADVANCING SPOKEN LANGUAGE UNDERSTANDING THROUGH ESPNET. Arora et al. In ICASSP 2022.
> [2] Two-Pass Low Latency End-to-End Spoken Language Understanding. Arora et al. In INTERSPEECH 2022.

---

### Official Review · Reviewer_dEWS · 2023-08-10

**Soundness:** 4

**Excitement:**

4: Strong: This paper deepens the understanding of some phenomenon or lowers the barriers to an existing research direction.

**Paper Topic And Main Contributions:**

This paper addresses the problem of erroneous ASR output that deteriorate the performance of SLU task. While it is common to encounter such problem with noisy prediction output from ASR model, especially with noise corrupted speech, it is also difficult to work on the robustness of the downstream SLU task. Interestingly, this work proposes a two-stage multi-grained contrastive learning framework to handle the noisy ASR output and has achieved improved results.

**Questions For The Authors:**

Mentioned previously, how the author derived to their proposed data augmentation? How does it affect the model if they are random and executed casually? Would curriculum learning help with increasing errors in the perturbation and what are the frequency of such augmentation used?

**Reasons To Accept:**

The work is inspired by FILIP to use the fine-grained contrastive learning to improve the representations of the downstream model. Besides, it is interesting and shown to be effective based on how the author uses such contrastive learning loss on corrupted text output. It is good that such technique appears to be easy to reproduce (but requires the understanding of the augmentation techniques used by the author).

Even though the pre-training of the proposed work requires additional computational cost, it is a plus that the final inference remains unchanged to the base model. As such, the model does have strong applicative uses.

The paper is comprehensive and the ablation work is sufficient with detailed analysis shown in the results and appendix. The visual plots also help to understand the improved representation of such technique.

**Reasons To Reject:**

The effectiveness of the proposed work seems to tie partly to the data augmentation technique. Noted on the sensitivity of the data augmentation method in appendix, I am just curious if such augmentation requires careful and thoughtful engineering? Does substitution work with random replacement?

**Reproducibility:**

3: Could reproduce the results with some difficulty. The settings of parameters are underspecified or subjectively determined; the training/evaluation data are not widely available.

**Reviewer Confidence:**

3: Pretty sure, but there's a chance I missed something. Although I have a good feel for this area in general, I did not carefully check the paper's details, e.g., the math, experimental design, or novelty.

---

> ### Author Rebuttal · Authors · 2023-08-29
>
> We express our gratitude for the insightful reviews you've provided. We value your input and would like to address your concerns as outlined below.
>
> **On data augmentation.**
>
> - Thank you for your valuable questions. First, data augmentation is one of the important techniques for MCLF.
> Generally, in some scenes, the performance of the model can be improved by using the method of contrastive learning. However, contrastive learning often requires more data to exert its benefits (e.g., [1][2]) Based on this consideration, we designed DA method starting from the construction form of WER. However, we want to clarify that the DA method we use is definitely not optimal. We can replace it with many existing excellent DA methods. We will discuss this part in the next version, thank you for your comments!
>
> **Questions**
>
> **Q:** How the authors derived to the proposed DA method?
>   - We start with the construction form of the ASR error. As can be seen in Line 210, the Word Error Rate to measure the percentage of incorrectly transcribed words consists of Inserted words, deleted words, and substitution words. Thus we design the Insertion DA, deletion DA, and substitution DA. The experimental results show the effectiveness of such DA methods.
>
> **Q:** How does DA affect the model if they are random and executed casually?
> - We provided DA ablation in Table 7 in the Appendix, where we show that all of the Insertion, Deletion, and Substitution DA methods are useful for the training of MCLF. Following your recommendation, we try one more random DA as a comparison under fine-grained contrastive learning setting. Note that except for the augmented data, we keep other implementation settings exactly the same for fair comparison. We run five times by different random seeds. Results show that 1 time we achieved a 0.2 improvement in ATIS (this is nearly the same as Insertion DA), and 1 time did not have any impact on performance, but 3 times, the loss convergence during pre-trained was significantly slower, and final finetuned models' results are 1% less than the Insertion DA method on average. We will discuss this issue in detail in the next version, and sincerely appreciate your suggestions.
>
> **Q:** Would curriculum learning help with increasing errors in the perturbation and what are the frequency of such augmentation used?
> - Curriculum learning could potentially help with increasing errors in the perturbation, as it would allow the model to first learn from less challenging examples and gradually progress towards harder, more perturbed examples. This strategy can help the model develop a more robust understanding of the data and prevent it from overfitting on any specific set of perturbations. We will discuss this issue in detail in the next version, and sincerely appreciate your suggestions.
>
> **References**
>
> [1] FILIP: Fine-grained Interactive Language-Image Pre-Training. Hou et al., 2021.
> [2] Efficient and effective passage search via contextualized late interaction over bert. Omar et al., 2020.

---

### Official Review · Reviewer_7euW · 2023-08-12

**Soundness:** 3

**Excitement:**

3: Ambivalent: It has merits (e.g., it reports state-of-the-art results, the idea is nice), but there are key weaknesses (e.g., it describes incremental work), and it can significantly benefit from another round of revision. However, I won't object to accepting it if my co-reviewers champion it.

**Missing References:**

Multi-grained alignment for SLU systems:
- Label-aware Multi-level Contrastive Learning for Cross-lingual Spoken Language Understanding; Shining Liang, Linjun Shou, Jian Pei, Ming Gong, W. Zuo, X. Zuo, and Daxin Jiang
- Cross-modal Transfer Learning via Multi-grained Alignment for End-to-End Spoken Language Understanding; Y Zhu, Z Wang, H Liu, P Wang, M Feng, M Chen, X He

**Paper Topic And Main Contributions:**

This paper address the problem of automatic speech recognition (ASR) errors for cascaded spoken language understanding (SLU) models that takes ASR transcripts as input. The authors propose MCLF, a two-stage multi-grained contrastive learning framework to enhance the robustness of SLU systems. At the first stage, the approach adopts multi-grained contrastive learning alignment in both sequence and token level as pretraining objective. Different data augmentation (DA) strategies are also introduced to generate more transcripts with a certain level of word error rate (WER). During finetuning, a joint training objective is introduced with various classification losses are proposed.

**Questions For The Authors:**

- Q1: What data is used during pretraining? How is augmented data used in constructing positive and negative pairs during pretraining?

**Reasons To Accept:**

- S1: Fine-grained pretraining objective along with DA strategies for more ASR-robust system.
- S2: The approach is proven to be consistently effective across different model backbones, and it outperforms all baselines of previous works.
- S3: Detailed analyses on the model robustness in visualization, pooling strategies, as well as error case study, demonstrates the effectiveness of the method.

**Reasons To Reject:**

- W1: The idea of multi-grained alignment for SLU models is not novel; There have been similar SLU works in applying multi-level contrastive learning methods in SLU systems (see missing reference).
- W2: The contribution of the paper is a little marginal, where most of the techniques are already commonly used in SLU systems: the data augmentation method is straightforward, and the the finetuning losses are simple combinations of existing classification losses.
- W3: More thorough analysis on the contribution of different components within the model such as data augmentation and loss combination of the method is lacking.

**Reproducibility:**

3: Could reproduce the results with some difficulty. The settings of parameters are underspecified or subjectively determined; the training/evaluation data are not widely available.

**Reviewer Confidence:**

3: Pretty sure, but there's a chance I missed something. Although I have a good feel for this area in general, I did not carefully check the paper's details, e.g., the math, experimental design, or novelty.

**Typos Grammar Style And Presentation Improvements:**

- P1[Figure 1]: The legend could be moved to somewhere else not covering the plot;
- P2: It would be better to have figures in pdf format rather than image format;
- P3[line 309]: they have -> and they have
- P4[381]: different -> the same?
- P5[448]: interactive -> interaction

---

> ### Author Rebuttal · Authors · 2023-08-29
>
> We appreciate your detailed feedback; addressing these suggestions has improved the quality of the paper. Below are some explanations about the remarks of your concerns.
>
> **On the concern about novelty and missing references.**
>
> - We appreciate your feedback, particularly the pointing out of the Multi-level contrastive learning-based SLU systems referenced in [1][2]. Their input will be incorporated into the next version of our study and we will create a clear distinction between our approach and theirs. Additionally, we will compare our work with other frameworks that incorporate multi-grained information.
>
> - It's worth noting that our method differs significantly from the ones mentioned in [1][2]. In [1], contrastive learning (CL) is employed for Cross-lingual SLU tasks, while our goal is to alleviate the ASR-Error problem. Furthermore, in their work, positive samples are drawn from different languages, whereas we generate our positive samples by imitating ASR error.
>
> - In [2], fine-grained contrastive learning is used between different modalities, namely speech and text. However, in our training optimization target, only a single text modality is used.
> Moreover, the concept of multi-grained or fine-grained can also be seen in other areas outside of SLU, such as the field of vision-language pre-train (FILIP, [3]) and the field of passage search (ColBERT, [4]).
>
> - We believe these significant distinctions and additional insights strengthen the validity of our approach and broaden its implications.
>
> **On the concern about marginal contribution.**
>
> - Thank you for your valuable feedback on our paper. We understand your concerns regarding the use of common techniques. However, we would to clarify that the originality of our work is not focused on inventing complex new techniques, but rather on creatively employing these known strategies within a completely novel framework, namely the Multi-Grained Contrastive Learning Framework (MCLF). This uniquely aids in enhancing spoken language understanding (SLU) systems that are prone to ASR errors.
> - The results shown in Table 1 clearly highlight significant system performance improvements following the application of MCLF within the aforementioned paradigm. We believe that these findings represent a meaningful and substantial contribution to the field.
> - Although the utilization of underlying basic data augmentation and fine-tuning techniques is not our first invention, we are the first to successfully apply these techniques in the multi-granularity ASR error SLU scenario. Our paper illustrates their effectiveness in different backbones, illustrating novel applications of these widely used processes.
> - Consequently, we respectfully disagree with the characterization of our work's contribution as being marginal. Instead, we assert that our work offers significant utility to the field.
>
> **On the concern about DA and Loss Ablation.**
> - The ablation of data augmentation can be seen in Table 7 in Appendix C.
> - The ablation of loss ablation can be seen in Table 1 in Section 3.3. We conduct experiments on four backbones (i.e., RoBERTa, BERT, ELECTRA, and TinyBERT). For each backbone, four different loss combinations are experimented for getting insights. Row 1 means the original PLM without any contrastive learning objective; Row 2 means the PLM is trained along with global contrastive learning loss; Row 3 means the PLM is trained along with the fine-grained contrastive learning loss; Row 4 means the PLM is trained with both global contrastive learning loss and fine-grained contrastive learning loss.
>
> **On the concern about implementation details.**
>
> **Q:** What data is used during pretraining?
> - During pre-training, we use the same task-specific data but the learning target is MLM. Details can be seen in Table 5 and Table 6 in Appendix A & Appendix B.
>
> **Q:** How is augmented data used in constructing positive and negative pairs during pretraining?
> - A: We apply contrastive learning over oracle and asr sentences by optimizing InfoNCE loss during pre-training, details can be seen in Line 142 - Line 156. The positive data are generated by data augmentation, the negative data are sampled from the in-batch data. For more details on InfoNEC loss, please refer to [5][6]. Thanks for the careful review, for more details, our implemented code is appended along with the submitted manuscript with detailed comments.
>
> **Typos**
>
> - Thank you for your careful reading! We will fix them in the revised version.
>
>
> **References**
>
> [1] Label-aware Multi-level Contrastive Learning for Cross-lingual Spoken Language Understanding. Liang et al., 2022.
> [2] Cross-modal Transfer Learning via Multi-grained Alignment for End-to-End Spoken Language Understanding. Zhu et al., 2022.
> [3] FILIP: Fine-grained Interactive Language-Image Pre-Training. Hou et al., 2021.
> [4] Efficient and effective passage search via contextualized late interaction over bert. Omar et al., 2020.
> [5] https://paperswithcode.com/method/infonce.
> [6] Representation Learning with Contrastive Predictive Coding. Oord et al., 2018.

---

### Meta-Review · Area_Chair_FsB1 · 2023-09-14

**Recommendation:** 3

**Metareview:**

This paper introduces MCLF, a two-stage, multi-grained contrastive learning framework aimed at improving the robustness of SLU systems. In its initial stage, the framework employs multi-grained contrastive learning alignment at both the sequence and token levels as a pre-training objective. Various data augmentation (DA) strategies are utilized to produce additional transcripts with a specific word error rate (WER). In the finetun-ing phase, a joint training objective is applied, and multiple classification losses are proposed. MCLF showcases its efficacy across four public benchmarks.

**Soundness Scores**: (3, 4, 4)
All reviewers agree that the experiments are methodologically sound. The original manuscript includes comprehensive experiments on four public benchmarks as well as an ablation study. Although a detailed error analysis was absent in the initial submission, the authors have subsequently addressed this in their rebuttal. Overall, the primary claims of this work are well supported.

**Excitement Scores**: (3, 4, 3)
All reviewers agree that the novelty and contributions as incremental, primarily because the techniques detailed have already been extensively employed in SLU tasks. In their rebuttal, the authors seem to acknowledge that their proposed method's performance might be comparable to, or only slightly better than, random chance data augmentation.

---

### Decision · Program_Chairs · 2023-10-07

**Decision:**

Accept-Findings

**Comment:**

This paper introduces MCLF, a two-stage, multi-grained contrastive learning framework aimed at improving the robustness of SLU systems. In its initial stage, the framework employs multi-grained contrastive learning alignment at both the sequence and token levels as a pre-training objective. Various data augmentation (DA) strategies are utilized to produce additional transcripts with a specific word error rate (WER). In the finetun-ing phase, a joint training objective is applied, and multiple classification losses are proposed. MCLF showcases its efficacy across four public benchmarks.

**Soundness Scores**: (3, 4, 4)
All reviewers agree that the experiments are methodologically sound. The original manuscript includes comprehensive experiments on four public benchmarks as well as an ablation study. Although a detailed error analysis was absent in the initial submission, the authors have subsequently addressed this in their rebuttal. Overall, the primary claims of this work are well supported.

**Excitement Scores**: (3, 4, 3)
All reviewers agree that the novelty and contributions as incremental, primarily because the techniques detailed have already been extensively employed in SLU tasks. In their rebuttal, the authors seem to acknowledge that their proposed method's performance might be comparable to, or only slightly better than, random chance data augmentation.